# SiT Dataset: Socially Interactive Pedestrian Trajectory Dataset for Social Navigation Robots

**Jongwook Bae**[*]   **Jungho Kim**[*]   **Junyong Yun**[*]   **Changwon Kang**[*]   **Jeongseon Choi**
**Chanhyeok Kim**   **Junho Lee**   **Jungwook Choi**   **Jun Won Choi**[†]

Hanyang University

{jwbae, junghokim, jyyun, changwonkang, jschoi, chkim, jhlee}@spa.hanyang.ac.kr
{choij, junwchoi}@hanyang.ac.kr

## Abstract

To ensure secure and dependable mobility in environments shared by humans and robots, social navigation robots should possess the capability to accurately perceive and predict the trajectories of nearby pedestrians. In this paper, we present a novel dataset of pedestrian trajectories, referred to as *Social Interactive Pedestrian Trajectory* (SiT) dataset, which can be used to train pedestrian detection, tracking, and trajectory prediction models needed to design social navigation robots. Our dataset includes sequential raw data captured by two 3D LiDARs and five cameras covering a 360-degree view, two *inertial measurement units* (IMUs), and *real-time kinematic positioning* (RTK), as well as annotations including 2D & 3D boxes, object classes, and object IDs. Thus far, various human trajectory datasets have been introduced to support the development of pedestrian motion forecasting models. Our SiT dataset differs from these datasets in the following three respects. First, whereas the pedestrian trajectory data in other datasets were obtained from static scenes, our data was collected while the robot navigated in a crowded environment, capturing human-robot interactive scenarios in motion. Second, unlike many autonomous driving datasets where pedestrians are usually at a distance from vehicles and found on pedestrian paths, our dataset offers a distinctive view of navigation robots interacting closely with humans in crowded settings. Third, our dataset has been carefully organized to facilitate the training and evaluation of end-to-end prediction models encompassing 3D detection, 3D multi-object tracking, and trajectory prediction. This design allows for an end-to-end unified modular approach across different tasks. We introduce a comprehensive benchmark for assessing models across all aforementioned tasks and present the performance of multiple baseline models as part of our evaluation. Our dataset provides a strong foundation for future research in pedestrian trajectory prediction, which could expedite the development of safe and agile social navigation robots. The SiT dataset, development kit, and trained models are publicly available at: https://spalaboratory.github.io/SiT/

## 1   Introduction

With advances in technology in a variety of fields, social navigation robots are emerging as versatile vehicles that can provide a variety of services, including last-mile delivery, street cleaning, manufacturing, and security patrols. These mobile robots need to move safely and smoothly while interacting with humans in shared ways without collisions. A popular method of collision avoidance is to create an occupancy grid map and represent the location of obstacles on the map. In complex and crowded

---

[*]Equal contribution.
[†]Corresponding author.

37th Conference on Neural Information Processing Systems (NeurIPS 2023) Track on Datasets and Benchmarks.

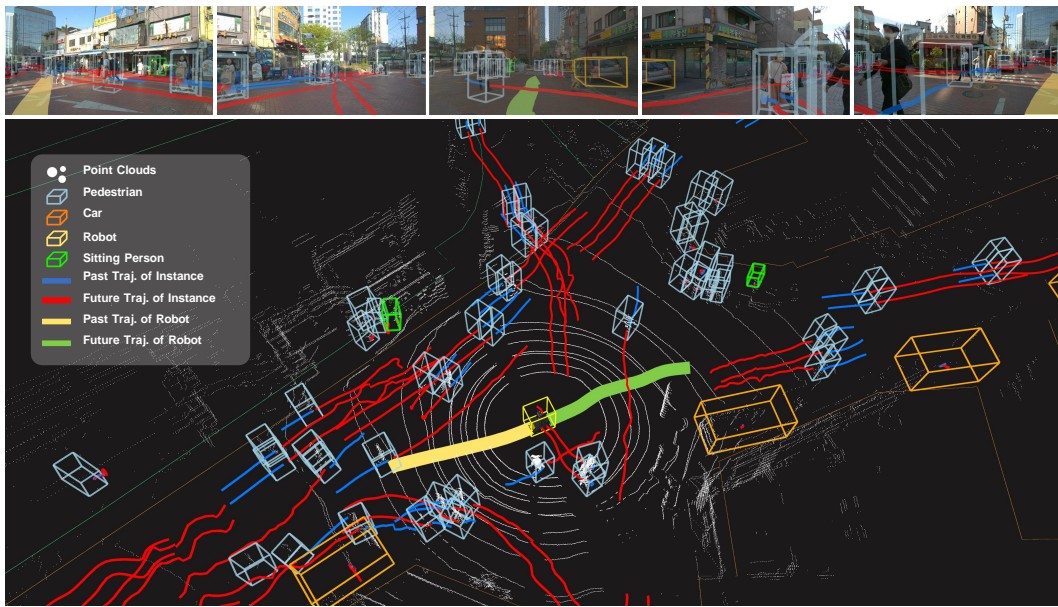

Figure 1: Visualization of pedestrian trajectory data collected from diverse social interactive environments in an outdoor scene (`Cafe_Street_3`). The first row showcases five different cameras, collectively capturing a full 360-degree view. In the lower section, point clouds, the mobile robot's trajectory, individual object trajectories, and their respective bounding boxes are displayed.

settings, traditional mobile robots often move at a slow pace and behave passively because they perceive people as obstacles and adopt conservative planning strategies to ensure a safe distance. For agile and safe navigation, robots need to perceive humans as individual entities in three-dimensional space, predict their movements, and utilize these predictions to avoid collisions, plan optimal paths, and navigate smoothly.

Recent advances in deep learning have led to significant improvements in performance for perceiving and predicting the motions of dynamic objects in diverse contexts. Several datasets have been made available to develop models designed specifically for predicting future human trajectories. While Argoverse [5, 32], nuScenes [4], and Waymo Open [27] datasets provide large-scale trajectory data capturing dynamic and interactive behaviors among agents, they were primarily designed within the context of autonomous driving scenarios, rather than for social navigation robots. ETH [22], UCY [16], and SDD [24] provide trajectories of pedestrians, mainly utilized for benchmarks as human trajectory prediction tasks. However, these trajectory datasets were collected only from static top-view scenes captured by cameras positioned at high vantage points such as rooftops, buildings, or drones hovering in the sky. Consequently, these datasets do not capture human-robot interactive scenarios. Furthermore, they do not provide raw sensor data with annotations required to train models for upstream tasks such as 3D object detection and multi-object tracking. STCrowd data [9] was collected from the sensors from a person-view perspective: however, the location of sensors also remained at a fixed position throughout the data collection process, resulting in a lack of variation in the scenes across different samples. Consequently, these datasets fail to capture the dynamic nature of human-robot interaction. JRDB [20] provides the trajectory data collected in interactive scenarios, but the data were not arranged in a trajectory form, and the multiple sensor data were not fully synchronized in time, limiting the potential of multi-sensor fusion.

Multiple studies in the field of *human-robot interaction* (HRI) have presented evidence revealing the impact of a robot's actions on pedestrians' walking behaviors [6, 7, 21, 29, 39]. We aim to expand the research area for pedestrian perception by introducing the HRI-included dataset that reflects the interactive behaviors between pedestrians and a mobile robot in the real world. In this paper, we present the *Social Interactive Pedestrian Trajectory* (SiT) dataset, which offers trajectories of pedestrians collected by a robot navigating in a diverse range of socially interactive scenarios as shown in Figure 1.

| Dataset | Platform | Sensor Data | Task | Sync. | Map | E2E | Location |
|---|---|---|---|---|---|---|---|
| UCY [16] | Fixed | Top-view Cam | T, P | - | | | Outdoor |
| ETH [22] | Fixed | Top-view Cam | T, P | - | | | Outdoor |
| SDD [24] | Fixed | Top-view Cam | T, P | - | | | Outdoor |
| CITR-DUT [35] | Fixed | Top-view Cam | T, P | - | | | Outdoor |
| nuScenes [4] | Vehicle | 360-view Cam, LiDAR, IMU, RTK | D, T, P | ✓ | ✓$^\dagger$ | | Outdoor |
| Waymo Open [27] | Vehicle | multi-view Cam, LiDAR, IMU, RTK | D, T, P | ✓ | ✓ | | Outdoor |
| Argoverse [5, 32] | Vehicle | 360-view Cam, LiDAR, IMU, RTK | D, T, P | ✓ | ✓$^\dagger$ | ✓ | Outdoor |
| JRDB [20] | Robot | 360-view Cam, LiDAR, IMU | D, T | | | | Indoor&Outdoor |
| STCrowd [9] | Fixed | Front-view Stereo Cam, LiDAR | D, T, P | ✓ | | | Outdoor |
| **SiT(Ours)** | **Robot** | **360-view Cam, LiDAR, IMU, RTK** | **D, T, P** | **✓** | **✓$^\dagger$** | **✓** | **Indoor&Outdoor** |

Table 1: Comparison of several pedestrian trajectory datasets. Dashes "-" represent attributes that are either inapplicable or unavailable. Sync. stands for Multi-Sensors synchronization. D, T, and P mean detection, tracking, and prediction, respectively. E2E indicates whether the dataset supports the joint design of upstream tasks. (†) includes multi-layered map information.

The SiT dataset consists of

1. Raw data acquired by two 3D scanning LiDARs, five cameras covering a 360-degree view, two *inertial measurement units* (IMUs), and *real-time kinematic positioning* (RTK)

2. 20 seconds of sequential data sampled at 10 Hz with indoor and outdoor scenes obtained from a mobile robot,

3. 9 seconds of trajectory data represented by vectors of poses, namely $(x, y, z)$

4. Annotations for 2D & 3D bounding boxes and object IDs

5. Ego-motion of the robot obtained by RTK (for outdoor settings) and the *simultaneous localization and mapping* (SLAM) algorithm (for indoor settings)

6. Multi-layered semantic maps obtained from LiDAR point clouds using the SLAM algorithm.

Our entire dataset comprises 60 scenes, totaling 60K images and 12K point cloud frames. This encompasses approximately 470K 2D annotations and 320K 3D annotations.

Table 1 highlights the distinctive features of the SiT dataset compared to existing human trajectory datasets. The main contributions of our dataset are summarized as follows.

- Our SiT dataset provides large-scale real-world pedestrian trajectory data obtained through a robot's navigation in densely populated indoor and outdoor environments, including the interiors of buildings, campuses, crosswalks, public pedestrian roads, and more.

- SiT dataset offers the flexibility to design trajectory prediction models using various contextual information including appearance features and ego-motion of a robot, and semantic map data, which could not be supported in the existing datasets.

- SiT dataset supports precise time synchronization between multi-modal sensors using a centralized sensor triggering method. This facilitates the development of efficient sensor fusion models tailored for perception and localization tasks.

- SiT dataset provides semantic map data for both indoor and outdoor scenes, encompassing multi-layered scene information. This comprehensive semantic map data can enhance the capability of motion prediction models that use static scene contexts around the robot.

- SiT dataset offers sequential raw data accompanied by 2D and 3D box annotations to support a joint modular design approach [12]. We have curated a benchmark for 3D object detection task, 3D *multi-object tracking* (MOT) task, and trajectory prediction task as well as an end-to-end prediction task that covers perception to motion forecasting.

- SiT dataset is publicly available. The SiT dataset has the potential to open up research opportunities in the development of learning-based 3D detection, 3D MOT, and trajectory prediction models in the context of social navigation robots.

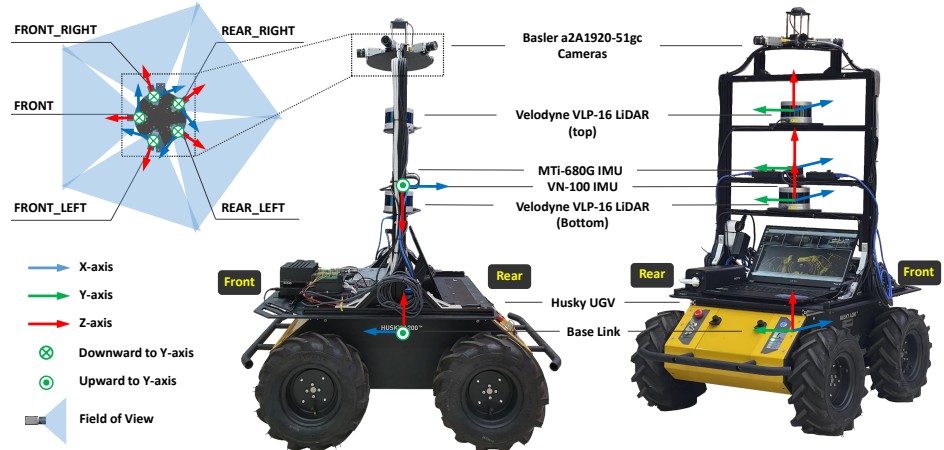

Figure 2: The Husky UGV platform [25] equipped with five cameras providing 360-view coverage, two 16-channel LiDARs, two IMUs, and RTK.

## 2 Related Work

ETH [22], UCY [16], and SDD [24] are popular pedestrian trajectory datasets that have been extensively utilized for developing pedestrian trajectory prediction models. These datasets were collected using static cameras positioned at elevated locations, capturing the $(x, y)$ coordinates that represent the spatial locations of pedestrians. Hence, these datasets do not specifically cater to the development of perception and motion prediction models for social navigation robots. Recently, STCrowd [9] has been introduced to facilitate the development of pedestrian perception models by providing camera and LiDAR sensor data. However, the viewpoint of these sensors was also fixed, which led to a limited variation in the scenes captured across different samples. Consequently, the aforementioned datasets fail to capture the dynamic nature of human-robot interaction. Argoverse [5, 32], nuScenes [4], and Waymo Open [27] datasets provide trajectories of vehicles and pedestrians collected using sensors installed on moving vehicles. These datasets primarily emphasize capturing vehicle-to-vehicle and vehicle-to-pedestrian interactions in driving scenarios. As a result, they may not be best suited to address the problems of social navigation robots. JRDB [20] offers camera and LiDAR sensor data collected from a mobile robot. However, it does not provide a benchmark specifically for the trajectory prediction task. Additionally, precise temporal synchronization between the camera and LiDAR data was not achieved in this dataset.

Table 1 provides a summary of the distinctive features offered by our SiT dataset in comparison to existing trajectory datasets. Our trajectory data is collected in scenarios where a robot navigates through crowded indoor and outdoor areas, actively interacting with pedestrians. The SiT dataset provides raw sensor data accompanied by annotations necessary for developing the models spanning from pedestrian perception to motion prediction. Furthermore, the SiT dataset achieves accurate time synchronization between different sensor modalities and provides semantic maps that describe the various scenes.

## 3 SiT Dataset

### 3.1 Robot Setup

We remotely operated Clearpath's Husky *unmanned ground vehicle* (UGV) platform [25] to collect different scenes of robot driving data. The robot has four large wheels that allow it to operate in a variety of environments. As shown in Figure 2, the robot was equipped with the following sensors:

- 2 x Velodyne VLP-16 rotating 3D LiDARs with 16 channels, 0.09 degree angular resolution, 30 degree vertical field of view, 2 cm distance accuracy, generating approximately 1.3 million points per second, and operating at 10 Hz.

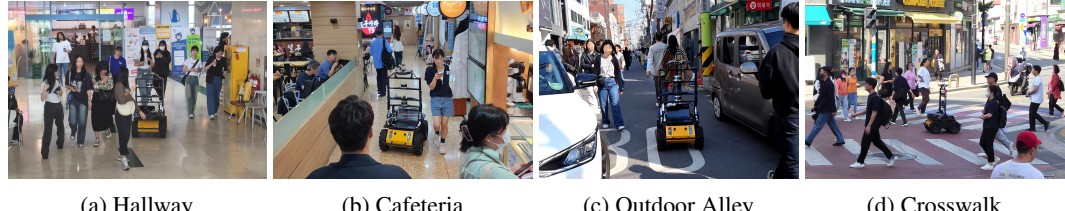

| (a) Hallway | (b) Cafeteria | (c) Outdoor Alley | (d) Crosswalk |

Figure 3: We have collected data from a variety of locations, both indoor and outdoor, that are densely populated with pedestrians.

- 5 x Basler a2A1920-51gc PRO GigE cameras capturing images at a high resolution of 1920 x 1200, a field of view of 54.3 degree (vertical) x 79 degree (horizontal), and covering the entire 360-degree area at 10 Hz.
- 1 x MTi-680G IMU & RTK offering pose and global positioning data at 20 Hz in 10 mm +1 ppm *circular error probability* (CEP), 0.5 degree RMS error of yaw, and 0.2 degree RMS error of roll and pitch.
- 1 x VectorNav VN-100 IMU providing pose data at 200 Hz with 1.0 degree RMS error of pitch and roll, and 2.0 degree RMS error of yaw.

Our mobile robot runs on Ubuntu 18.04 and *robot operating system* (ROS) Melodic. The data captured by the aforementioned sensors were recorded in the rosbag file format. The sensor data is available in both rosbag and raw data formats. Additionally, both intrinsic and extrinsic calibration parameters are included.

Previous studies have demonstrated that 3D object detection models based on multiple sensors produce more robust detection results than single sensor-based methods [2, 8, 17, 28, 33, 37]. Sensor fusion achieves maximum benefits when the camera and LiDAR sensors are precisely synchronized in time. To ensure this synchronization, we employed a *pulse per second* (PPS) signal generator to trigger two 3D LiDARs and five cameras. This method enables us to obtain accurately synchronized data from multiple cameras and LiDARs.

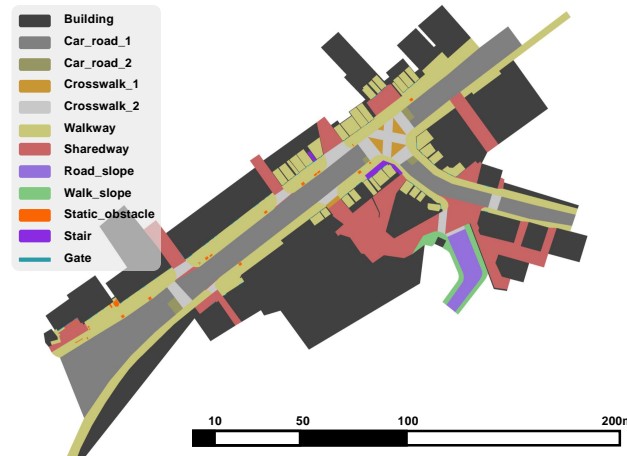

Figure 4: An example of a 12-layered semantic map from the outdoor scene. Distinct colors are used to represent different layers.

## 3.2 Data Collection

The data was collected in the downtown area of Seoul, South Korea. We selected a variety of indoor and outdoor locations including hallways, cafeterias, university buildings, streets, and crosswalks, as shown in the examples in Figure 3. Our dataset contains a diverse and representative set of real-world driving scenarios. We manually operated a robot platform throughout densely populated areas. We directed the robot to adhere to social norms when interacting with pedestrians, ensuring it avoided collisions and respected personal space.

## 3.3 Robot Localization

The robot's pose needs to be determined to compensate ego-motion for trajectory generation and utilize semantic maps for the motion prediction task. We used different localization methods for indoor and outdoor scenes. We used RTK to estimate the pose of the robot in outdoor scenes. We implemented the LiDAR-inertial SLAM algorithm as described in [26] for indoor scenes. By leveraging these localization techniques, we obtained accurate absolute location information for pedestrians, facilitating object tracking and prediction tasks.

### 3.4 Semantic Map

Using scene context information has been proven beneficial in enhancing trajectory prediction models [13, 15, 19]. To facilitate this, the SiT dataset offers multi-layered semantic maps that encompass a wide range of scene information. The semantic maps were generated through a two-step process. Initially, point cloud maps were constructed, followed by manual segmentation using the *ASSURE mapping tools* [10]. As shown in Figure 4, these maps use a twelve-layer hierarchical structure categorized by varying levels of detail.

### 3.5 Annotations

The SiT dataset contains annotations consisting of 2D bounding boxes of images captured by multi-view cameras covering 360-degree, and 3D cuboids expressed in 3D world coordinates. The 2D boxes were generated based on the existing 3D cuboids sharing object IDs. While our primary focus is to offer a pedestrian trajectory dataset, we have broadened the annotated classes to include car, bus, truck, cyclist, and motorcyclist. The distribution of class categories in our dataset is presented in Figure 5. To maintain a high standard of annotations, frames taken at a 5Hz sampling rate were manually labeled by expert annotators. Subsequently, the labels were interpolated to a higher frequency of 10Hz. This detailed granularity is crucial to meet the real-time requirements of perception and prediction in navigation robots.

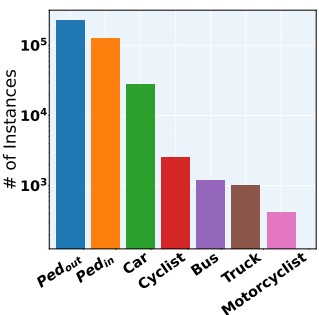

Figure 5: Histogram of 3D cuboids annotated for each class of objects present in the SiT Dataset.

Trajectory samples were produced by compensating for the ego-motion in the pedestrian tracks observed over a certain duration.

### 3.6 Structure of Dataset

The SiT dataset was split into training, validation, and test subsets. We split the dataset so that each subset encompasses different scenes. The distribution of pedestrian distances from the robot and their velocities are shown in Figure 6. Consistent pedestrian behaviors can be seen across the different subsets. Our dataset has been processed to facilitate the training of comprehensive end-to-end models covering multiple tasks [12].

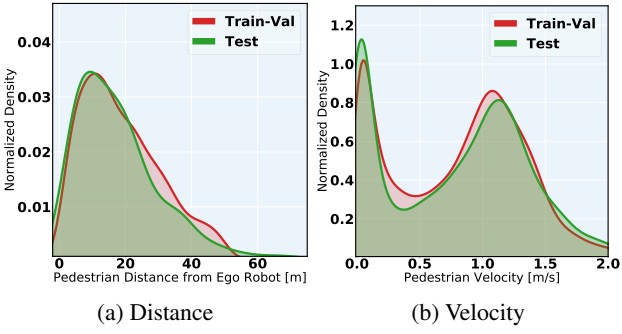

(a) Distance         (b) Velocity

Figure 6: Distribution of distance and velocity for Training-Valid and Test subsets.

### 3.7 Privacy

To prioritize privacy, all images have been processed to blur pedestrian faces and license plates. This careful curating ensures the responsible use of the data while maintaining the integrity of the information for research purposes. Detailed information on privacy concerns can be found in the Supplementary Material.

## 4 Data Analysis

In this section, we analyze the characteristics exhibited by the trajectory data in the SiT dataset. Figure 7 shows the distribution of pedestrians' locations surrounding the robot in our dataset, where $(0,0)$ denotes the location of the robot or ego-vehicles. To provide a basis for comparison, the distributions from two widely used self-driving datasets, Waymo Open [27] and nuScenes [4] datasets are also included. In the SiT dataset, pedestrians are predominantly located closer to the robot compared to other autonomous driving datasets. This can be attributed to the fact that both the robot and pedestrians share the same path or area. In contrast, in the Waymo Open and nuScenes datasets,

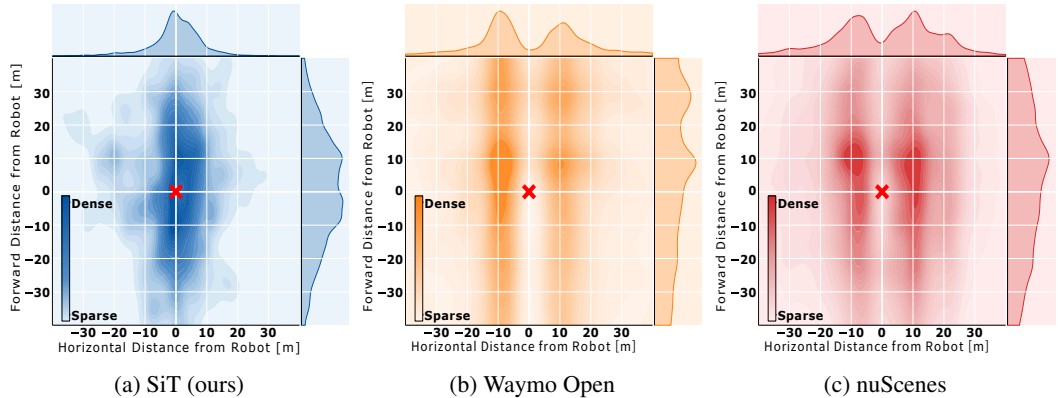

| | | |
|---|---|---|
| (a) SiT (ours) | (b) Waymo Open | (c) nuScenes |

Figure 7: Comparison of the spatial distribution of pedestrians relative to the ego-vehicle across the SiT, Waymo Open, and nuScenes datasets using *kernel density estimates* (KDEs). The red cross mark on the origin point (0, 0) in each plot is designated as the center of the robot or ego-vehicle.

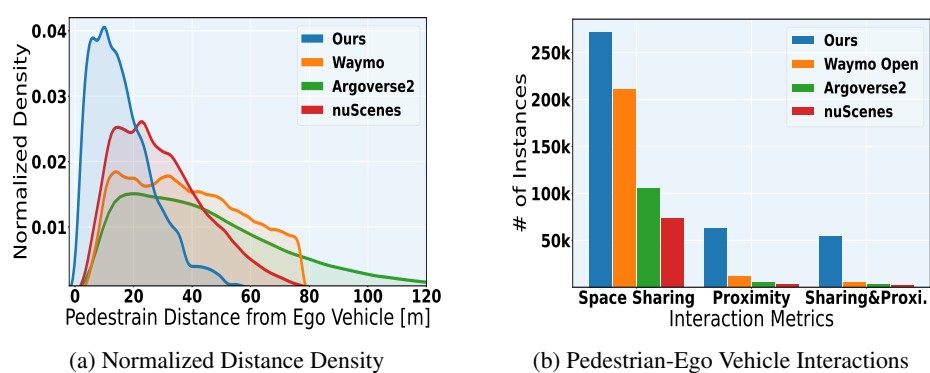

| | |
|---|---|
| (a) Normalized Distance Density | (b) Pedestrian-Ego Vehicle Interactions |

Figure 8: Comparison with other autonomous vehicle datasets. (a) represents the normalized density graph of the distance between the robot (or ego-vehicle) and surrounding pedestrians, while (b) illustrates the number of instances that satisfy some criteria. **Space sharing** condition refers to instances where the robot and pedestrians occupy the same semantic space, **Proximity** condition indicates cases where the distance between the robot and pedestrian is within 2 meters, and **Sharing & Proxi.** indicates cases that fulfill both conditions.

ego-vehicles and pedestrians typically occupy separate roads, resulting in fewer close interactions. We observe that pedestrians are often positioned alongside ego-vehicles, rather than directly in front or behind them. Conversely, in our dataset, pedestrians can be found surrounding the robot from all angles.

Figure 8 demonstrates other attributes of the SiT dataset. Figure 8a shows the distribution of the distance of pedestrians from the robot. This confirms that in our dataset, the robot engaged in close-distance interactions with pedestrians. Figure 8b compares the number of trajectory instances that satisfy two conditions; 1) *space sharing* and 2) *proximity*. *Space sharing* condition indicates cases where the robot and pedestrian occupy the same semantic domain. We check this condition by examining the areas of the semantic map where the robot and pedestrians are located. Typically, when the robot traverses a roadway while a pedestrian walks along a sidewalk, their behaviors tend to remain unaffected by each other due to the physical separation of their paths. On the other hand, interactive scenarios arise when they occupy the same semantic space. *Proximity* condition refers to cases where the distance between the robot and pedestrian is within 2 meters, indicating close physical proximity. Figure 8b shows that the SiT dataset exhibits a significantly higher number of instances that satisfy these two conditions compared to the other three autonomous driving datasets. This indicates that trajectory datasets designed for autonomous driving may not adequately represent the human-robot interactions for social navigation robots.

| Methods | Modality | mAP ↑ | AP(0.25) ↑ | AP(0.5) ↑ | AP(1.0) ↑ | AP(2.0) ↑ |
|---|---|---|---|---|---|---|
| FCOS3D [30] | Camera | 0.244 | 0.024 | 0.159 | 0.329 | 0.463 |
| PointPillars [14] | LiDAR | 0.351 | 0.260 | 0.354 | 0.374 | 0.418 |
| Centerpoint-P [36] | LiDAR | 0.414 | 0.300 | 0.424 | 0.446 | 0.486 |
| Centerpoint-V [36] | LiDAR | 0.518 | **0.397** | 0.531 | 0.553 | 0.592 |
| TransFusion-P [2] | LiDAR+Camera | 0.390 | 0.248 | 0.371 | 0.437 | 0.507 |
| TransFusion-V [2] | LiDAR+Camera | **0.531** | 0.318 | **0.536** | **0.607** | **0.665** |

Table 2: Evaluation of 3D pedestrian detection baselines. mAP is obtained from the average of AP (0.25), AP (0.5), AP (1.0), and AP (2.0). ↑ : the higher, the better.

| Methods | sAMOTA ↑ | AMOTA ↑ | AMOTP(m) ↓ | MOTA ↑ | MOTP(m) ↓ | IDS ↓ |
|---|---|---|---|---|---|---|
| PointPillars [14] + AB3DMOT [31] | 0.4110 | 0.1047 | 0.3580 | 0.4086 | 1.0277 | 1048 |
| Centerpoint Detector [36] + AB3DMOT [31] | 0.4841 | 0.1398 | 0.3958 | 0.4586 | 0.9836 | **554** |
| Centerpoint Tracker [36] | **0.6070** | **0.2007** | **0.2679** | **0.4760** | **0.5140** | 1136 |

Table 3: Evaluation of 3D pedestrian tracking baselines. ↑ : the higher, the better. ↓: the lower, the better.

## 5 Benchmarks and Experiments

In this section, we introduce the benchmarks designed for evaluating various 3D perception and motion prediction models. We also present the performance results of several baseline models on these benchmarks.

### 5.1 Benchmarks

Detailed information on the computation of these performance metrics can be found in the Supplementary Material.

**3D Pedestrian Detection.** The SiT dataset offers 3D pedestrian detection benchmark. The models generate 3D cuboids that encompass pedestrians in the scene for a given input frame. To evaluate the performance of 3D object detectors, *mean average precision* (mAP) metric is widely used. The success of detection can be determined based on different criteria. For example, KITTI dataset [11] used *intersection over Union* (IoU) metric while nuScenes dataset [4] used the distance between two cuboid centers. In the SiT dataset, we adopt AP based on the distance metric using the thresholds of $0.25$, $0.5$, $1.0$, and $2.0m$.

**3D Pedestrian Tracking.** The SiT dataset offers the benchmark for evaluating 3D pedestrian tracking algorithms. Given a sequence of data frames, the 3D pedestrian tracking models generate object tracks that indicate the pedestrians' movements over time. We adopt widely used performance metrics including sAMOTA, AMOTA, AMOTP, MOTA, MOTP, and *identity switches* (IDS) [3, 31] to evaluate the performance of tracking models.

**Pedestrian Trajectory Prediction.** The pedestrian trajectory prediction benchmark is presented to evaluate the accuracy of trajectory prediction. *Average displacement error* (ADE) and *final displacement error* (FDE) are widely used as evaluation metrics [1, 22]. ADE represents the average error in predicted trajectories compared to ground truth over all time steps, and FDE quantifies the deviation between the predicted final position and ground truth at the last time step. Using the preceding 2-second trajectory as input, the trajectory prediction models generate a set of $K$ 7-second future trajectories, representing $K$ distinct modes of future trajectory distribution. To evaluate the prediction accuracy of these $K$ trajectories, we use $\mathrm{ADE}_K$ and $\mathrm{FDE}_K$, which represent the minimum ADE and FDE values among the $K$ trajectory candidates.

**End-to-End Motion Prediction.** In the end-to-end motion prediction task, the models take raw sensor data as input and generate future 3D bounding boxes, accompanied by the corresponding object trajectories. We consider a 7-second future horizon for trajectory prediction. We use $\mathrm{mAP}_f$ metric proposed in FutureDet [23] along with the conventional ADE and FDE as performance metrics.

| Methods | Map | $ADE_5 \downarrow$ | $FDE_5 \downarrow$ | $ADE_{20} \downarrow$ | $FDE_{20} \downarrow$ |
|---|---|---|---|---|---|
| Social-LSTM [1] | | 1.638 | 3.121 | 1.630 | 3.103 |
| Y-Net [19] | | 1.527 | 2.802 | 0.836 | 1.878 |
| Y-Net [19] | ✓ | 1.361 | 2.624 | 0.675 | 1.547 |
| NSP-SFM [38] | | 1.346 | 2.261 | 0.634 | 1.087 |
| NSP-SFM [38] | ✓ | **1.061** | **1.818** | **0.517** | **0.925** |

Table 4: Evaluation of pedestrian trajectory prediction baselines based on $ADE_5$, $FDE_5$, $ADE_{20}$, and $FDE_{20}$. $\downarrow$: the lower, the better.

| Methods | $mAP \uparrow$ | $mAP_f \uparrow$ | $ADE_5 \downarrow$ | $FDE_5 \downarrow$ |
|---|---|---|---|---|
| FaF [18] | **0.490** | **0.079** | **1.915** | **3.273** |
| FutureDet-P [23] | 0.209 | 0.037 | 2.532 | 4.537 |
| FutureDet-V [23] | 0.408 | 0.053 | 2.416 | 4.409 |

Table 5: Evaluation of end-to-end motion prediction baselines. $\uparrow$: the higher, the better. $\downarrow$: the lower, the better.

## 5.2 Performance of Baseline Models

In this section, we present the performance of several baseline models for each benchmark. All experiments were performed on 4 NVIDIA RTX3090 GPUs with 2 Intel Xeon CPUs. Detailed configurations of the baseline models can be found in the Supplementary Material.

**3D Pedestrian Detection.** Table 2 presents the performance of four baseline methods, FCOS3D [30], PointPillars [14], CenterPoint [36], and TransFusion [2] evaluated in 3D Pedestrian Detection benchmark. CenterPoint-P and TransFusion-P utilized the PointPillars as encoding backbone, while CenterPoint-V and TransFusion-V employed SECOND [34] as the voxel encoding backbone. Note that camera-LiDAR fusion yields better performance compared to models relying on a single modality.

**3D Pedestrian Tracking.** We considered AB3DMOT [31] and the CenterPoint tracker as the baseline models for 3D MOT. We applied AB3DMOT to PointPillar and CenterPoint detectors. Table 3 presents the performance of the baselines evaluated on the SiT dataset.

**Pedestrian Trajectory Prediction.** Table 4 presents the performance of Social-LSTM [1], Y-Net [19], and NSP-SFM [38]. For Y-Net and NSP-SFM, the scene encoding was conducted utilizing the semantic maps offered from our dataset. In Table 4, we confirm the benefit of utilizing scene contexts for trajectory prediction.

**End-to-end Motion Prediction.** Table 5 presents the performance of two end-to-end motion prediction baseline models: Fast and Furious [18] and FutureDet [23]. FutureDet-P uses PointPillars as the encoding backbone while FutureDet-V utilizes SECOND as the voxel encoding backbone.

## 6 Conclusions

In this paper, we introduced the SiT dataset, a new pedestrian trajectory dataset designed to facilitate the development of perception and motion prediction models for social navigation robots. The SiT dataset comprises pedestrian trajectories gathered from a variety of social interaction scenarios that can be encountered by robots during navigation in the real world. Our analysis revealed that the SiT dataset effectively captured the behavior of pedestrians in close proximity to the robot within a shared semantic space on the map. The SiT dataset also offered high-quality annotations for 2D and 3D object detection and tracking, enabling the design of end-to-end motion prediction models in the upstream pipeline. These features make the SiT dataset unique when compared to existing pedestrian trajectory datasets. We hope that our proposed dataset and baselines provide a strong foundation for future research in pedestrian perception and have the potential to accelerate the development of fully autonomous robots.

One limitation of our dataset is that we were unable to scale the size of the dataset up to hundreds of thousands of frames due to time constraints. However, we are planning to address this limitation by adding an additional dataset in the second phase release, called *SiT dataset v2*, to significantly expand the scale of our dataset.

## Acknowledgment

This work was partly supported by 1) National Research Foundation of Korea(NRF) grant funded by the Korea government(MSIT) (No.2020R1A2C2012146), 2) Institute of Information & communications Technology Planning & Evaluation (IITP) grant funded by the Korea government(MSIT) (No.2020-0-01373, Artificial Intelligence Graduate School Program (Hanyang University) and 3) No.2022-0-00957, Distributed on-chip memory-processor model PIM (Processor in Memory) semiconductor technology development for edge applications), and 4) Hyundai Motor Company.

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
