# Supplementary Material

The Supplementary Material is organized as follows: Section A provides additional information on the SiT dataset. Section B provides the implementation details of the perception models. In Section C, we introduce the performance metrics used for the benchmarks.

## A    Additional Details for SiT Dataset

In this section, we discuss the details of the SiT dataset.

### A.1    Additional Statistics of SiT Dataset

Figure 1 presents the distribution of the distance from the robot for different object categories. Figure 2 presents the distribution of object velocity. Figure 3 compares the distribution of pedestrian position between outdoor and indoor environments. In outdoor settings, the spatial distribution of pedestrian position from the robot is broader in both longitudinal and lateral directions than in indoor scenes. This can be attributed to the expansive nature of outdoor environments. Figure 4 illustrates the distribution of distance from the robot and velocity of pedestrians in both indoor and outdoor environments. Figure 5 presents the distribution of 3D and 2D box sizes for pedestrians, as well as the yaw angle of 3D pedestrian boxes.

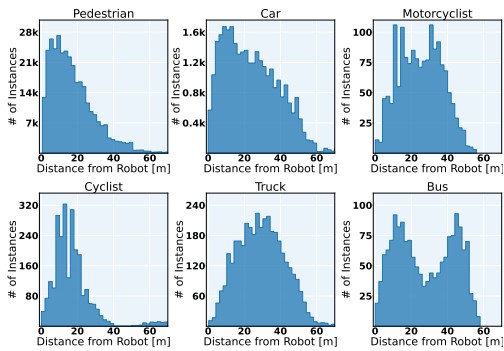

Figure 1: Distribution of distance from the robot for different object classes.

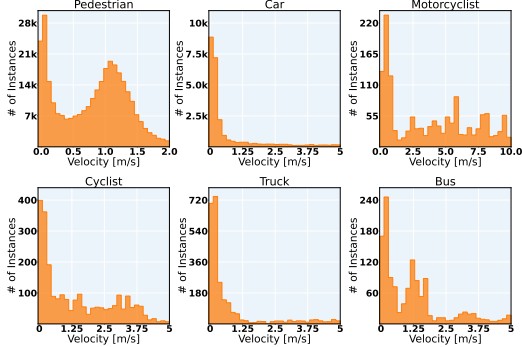

Figure 2: Distribution of velocity for different object classes.

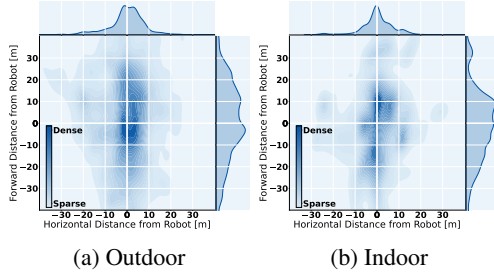

(a) Outdoor    (b) Indoor

Figure 3: Distribution of pedestrian positions in both outdoor and indoor environments, where $(0,0)$ indicates the position of the robot.

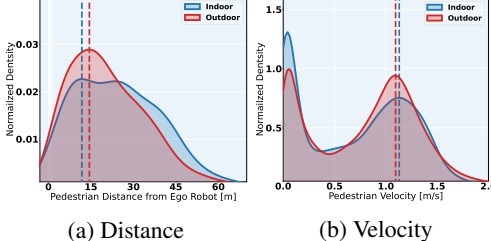

(a) Distance    (b) Velocity

Figure 4: Distribution of distance from the robot and velocity of pedestrians in both indoor and outdoor environments.

### A.2    Additional Examples of SiT Dataset

Figure 6 presents an illustration of more data samples in the SiT dataset.

### A.3    Interaction Examples of SiT Dataset

Our dataset is intended to capture natural and unbiased interactions between pedestrians and the robot in real-world scenarios. We avoided controlled setups and did not provide any guidance to

participants, collecting trajectory data from pedestrians in both indoor and outdoor environments. Figure 7 provides several examples of trajectories of the robot and pedestrians capturing real-world interactions like approaching, following, and collision avoidance.

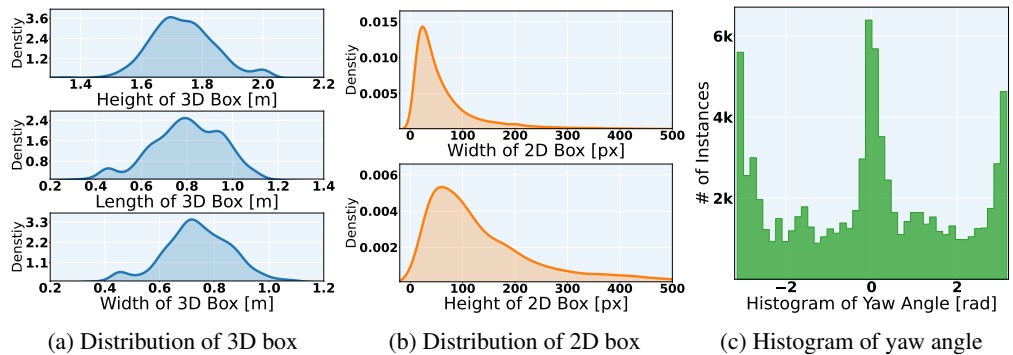

(a) Distribution of 3D box        (b) Distribution of 2D box        (c) Histogram of yaw angle

Figure 5: Distribution of size of both 3D and 2D pedestrian boxes as well as yaw angle of 3D pedestrian boxes.

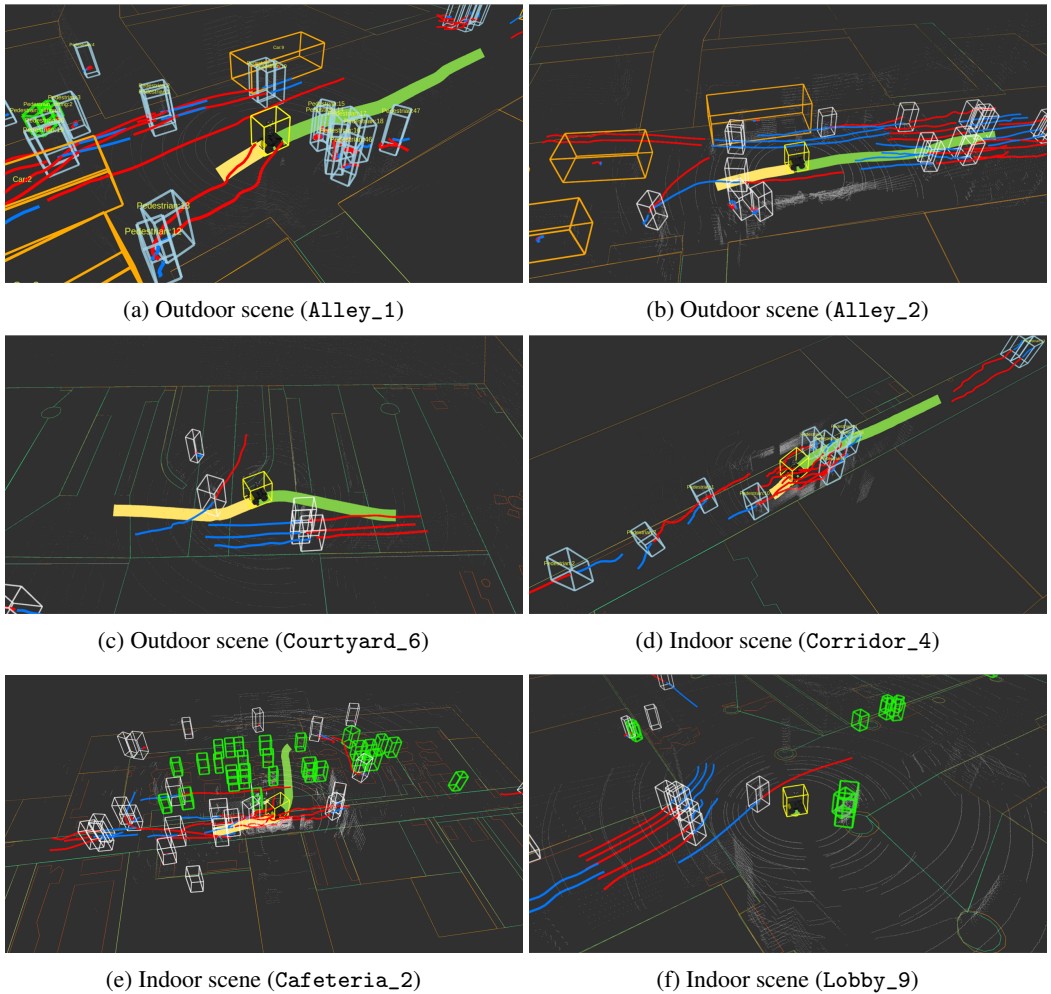

(a) Outdoor scene (`Alley_1`)

(b) Outdoor scene (`Alley_2`)

(c) Outdoor scene (`Courtyard_6`)

(d) Indoor scene (`Corridor_4`)

(e) Indoor scene (`Cafeteria_2`)

(f) Indoor scene (`Lobby_9`)

Figure 6: Illustration of additional examples of SiT dataset.

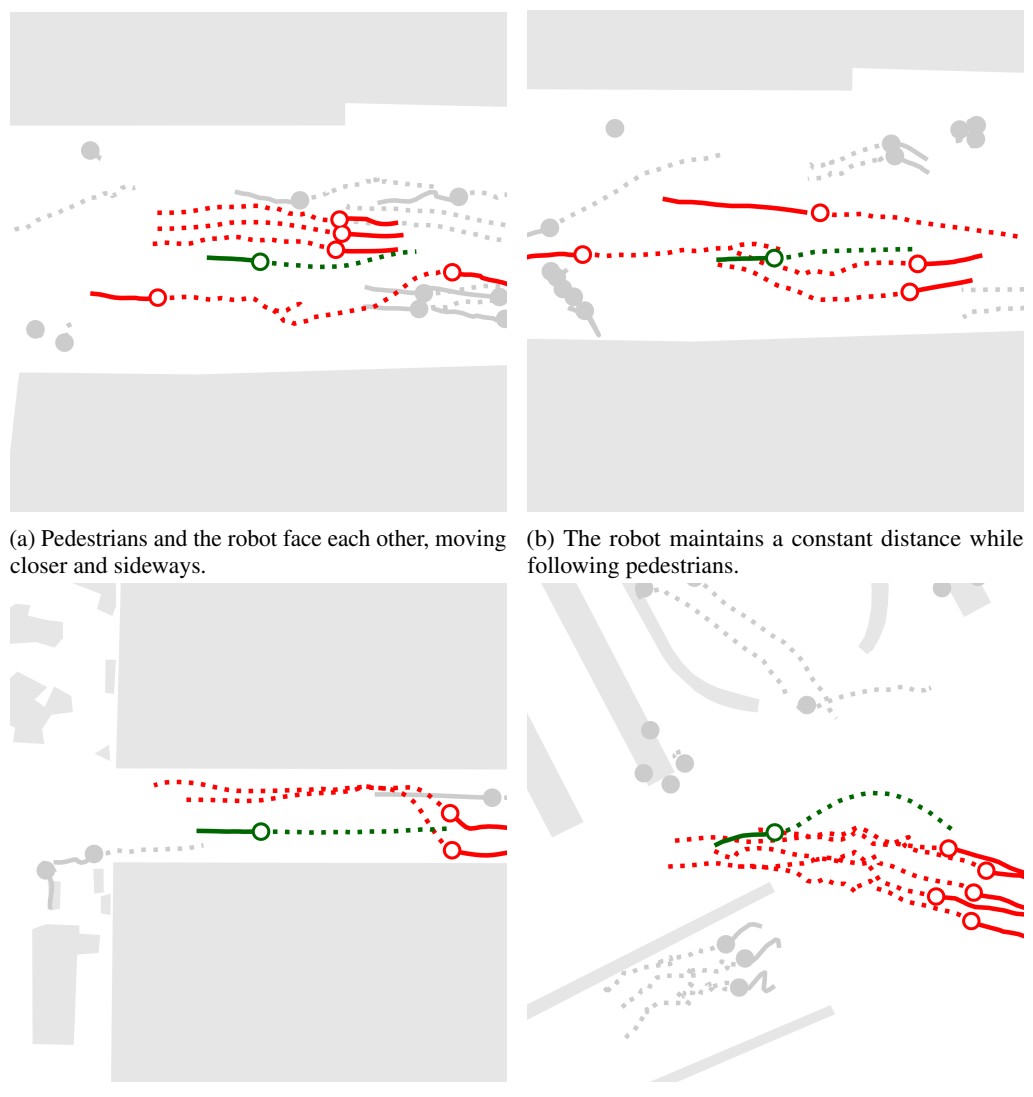

(a) Pedestrians and the robot face each other, moving closer and sideways.

(b) The robot maintains a constant distance while following pedestrians.

(c) Pedestrians avoid collision with the robot.

(d) The robot avoids collision with pedestrians.

Figure 7: Examples of human-robot interactions in the SiT dataset. Dotted lines, solid lines, and circles indicate future trajectories, past trajectories, and current positions, respectively. The robot, the highlighted pedestrians, and other objects are represented by green, red, and dark gray colors, respectively. The background map is shown in light gray.

## A.4    Additional Information on Semantic Map

Figure 8 presents examples of semantic maps included in the SiT dataset. In Table 1, we describe the attributes of each layer used for the semantic maps.

## A.5    Additional Information on Privacy Concerns

All data instances in the SiT dataset are carefully processed to remove identifiable information from the images. We identified faces using RetinaNet [7] and blurred them. For license plates, we labeled a portion of our image data with license plates and then trained Faster-RCNN [5] to blur them. After blurring the faces and license plates with the models, we checked each frame for additional blur to ensure that latent identifiers such as faces in the image dataset are completely anonymized, as shown in Figure 9.

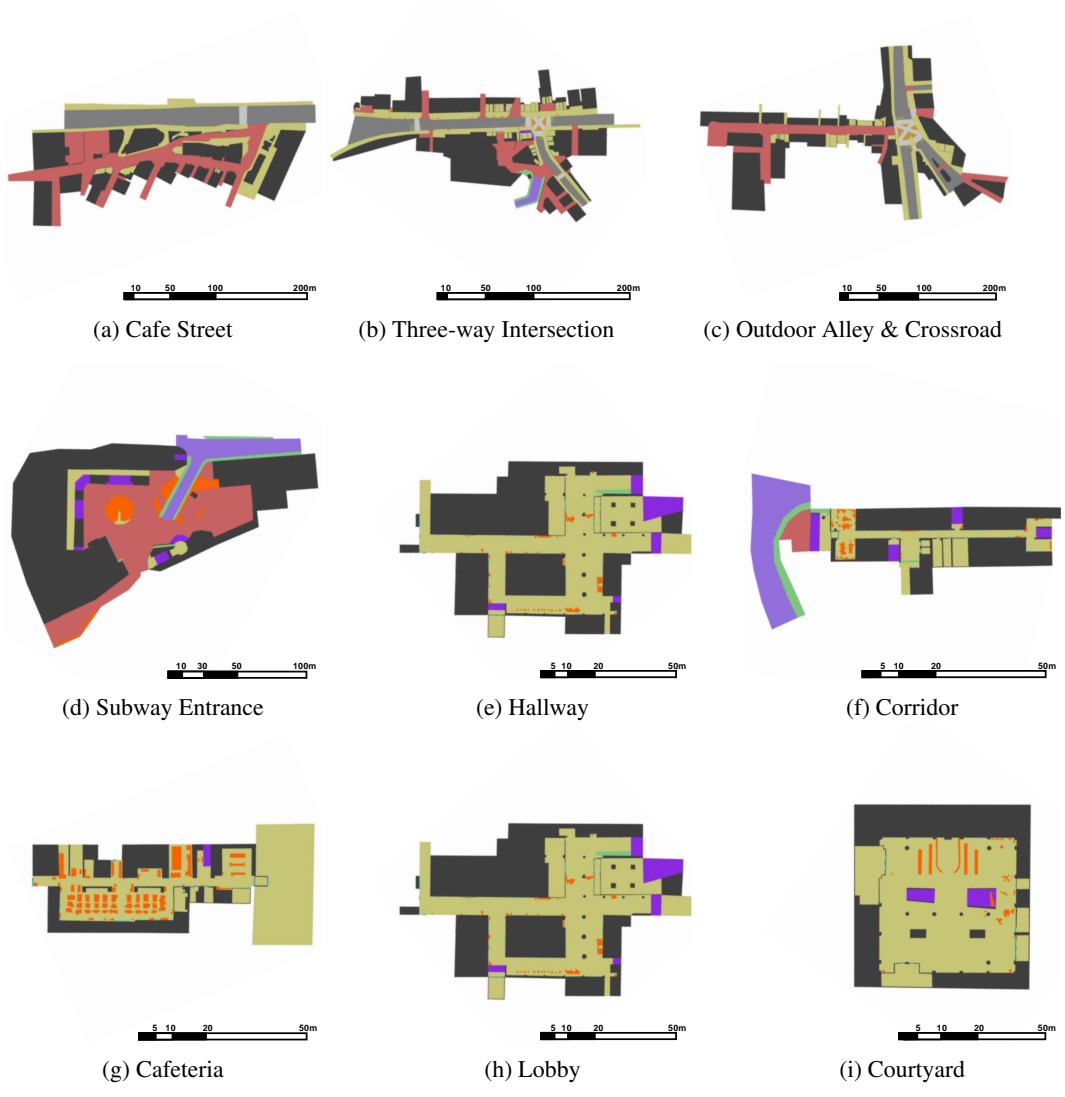

Figure 8: Illustration of semantic maps obtained from the SiT dataset.

### A.6 Licence

The SiT dataset is published under the CC BY-NC-ND License 4.0, and all codes are published under the Apache License 2.0.

## B Implementation details

In this section, we provide additional details on the 3D pedestrian detection, 3D *Multi-Object Tracking* (MOT), pedestrian trajectory prediction, and end-to-end pedestrian motion forecasting baselines. Our experiments were conducted on systems running Ubuntu 18.04, equipped with 2 Intel Xeon CPUs and 4 NVIDIA RTX3090 GPUs. Each experiment was performed three times with three distinct seeds, and the median of these results was reported as the experimental results in the body of the paper. Figure 10 presents the error bar plots for the 3D pedestrian detection, trajectory prediction, and end-to-end pedestrian motion forecasting experiments.

| Attribution | Decimal Code (R,G,B) | Type | Description |
|---|---|---|---|
| Building | (64, 64, 64) | | Impassable areas in structures such as buildings |
| Car_road1 | (128, 128, 128) | | Vehicle-only areas generally inaccessible to pedestrians |
| Car_road2 | (150, 150, 100) | | Areas between stop lines and crosswalk |
| Crosswalk_1 | (200, 150, 50) | | Area encompassing crosswalk denoting the overall designated crossing zone |
| Crosswalk_2 | (200, 200, 200) | | Areas of the crosswalks |
| Walkaway | (200, 200, 120) | 2D POLYGONS | Areas exclusively for pedestrian use |
| Sharedway | (200, 100, 100) | | Areas accommodating both pedestrians and vehicles |
| Road_slope | (147, 112, 219) | | Areas including slopes in roads |
| Walk_slope | (128, 200, 128) | | Areas including slopes within walkaway |
| Static_obstacle | (250, 100, 1) | | Areas of static obstacles including trees, tables, bollards, etc |
| Stair | (138, 43, 226) | | Areas of stairways used by pedestrians |
| Gate | (1, 154, 205) | 2D POLYLINE | Lines for entrances allowing pedestrians to pass through buildings |

Table 1: Definition of each layer from semantic map

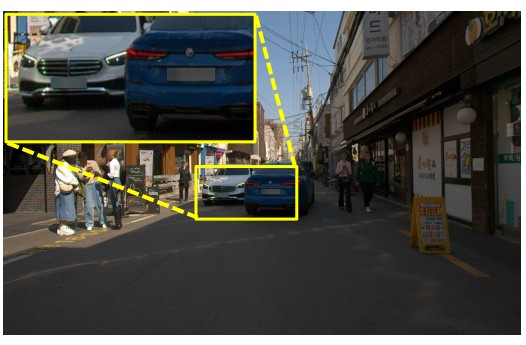 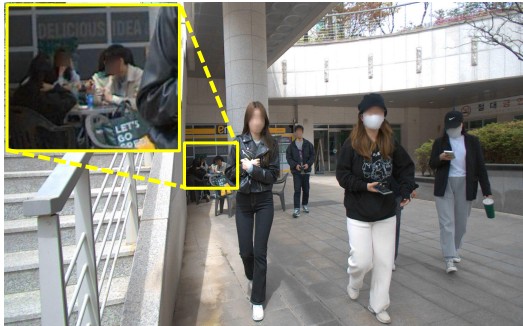

(a) Blurred license plates in (Outdoor_Alley_1)  (b) Blurred faces in (Courtyard_8)

Figure 9: Examples of images with blurred identifiable information such as license plates and pedestrian faces.

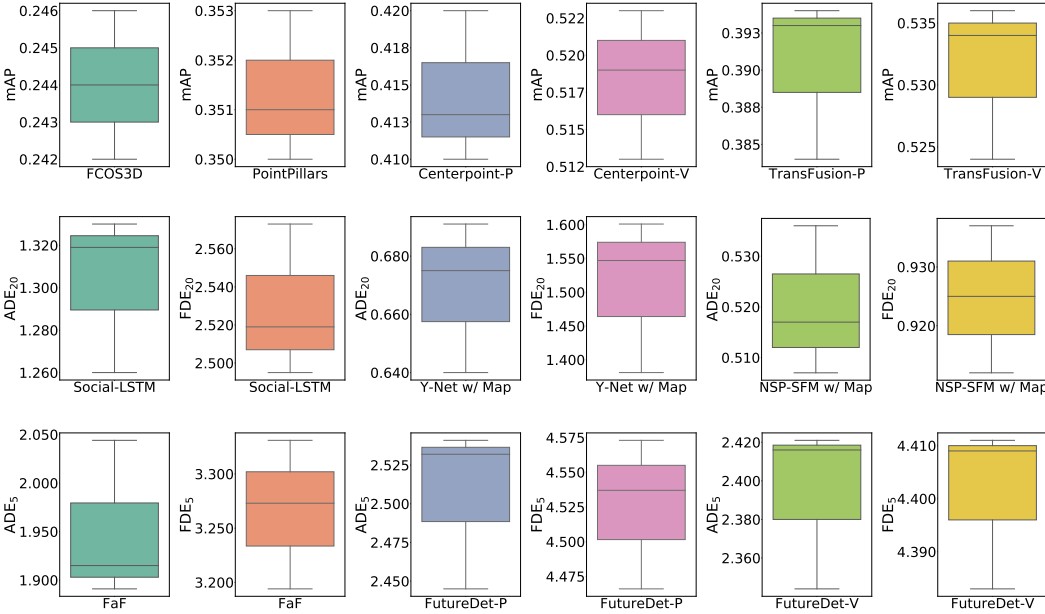

Figure 10: Error bar plots for the implemented models.

### B.1 3D Pedestrian Detection Models

We implemented FCOS3D [11], PointPillars [6], CenterPoint [13] and Transfusion [2] using the PyTorch 1.7.1 framework and the 1.0 version of MMDetection3D [4]. Employing the AdamW optimizer with a learning rate of 0.001, we trained each model for 20 epochs. The batch sizes were configured to 4, 16, 24, and 8 respectively. We resized the image by half for the training of the Transfusion.

### B.2 3D Multi-Object Tracking Models

Based on the detection experiment results of the PointPillars and CenterPoint(V) models, we implemented AB3DMOT [12] and Centerpoint tracker for the 3D MOT task. For AB3DMOT, the threshold for box association is 2 meters based on the 3D distance metric. For Centerpoint tracker, we used the Hungarian algorithm with a maximum age set to 3. The remaining settings for both AB3DMOT and the Centerpoint tracker follow the default settings in the author code. For the evaluation, we used the official nuScenes evaluation code [3], which uses a distance-based evaluation method. The distance threshold was set to 2 meters and we used 41 recall values from 0 to 1.

### B.3 Pedestrian Trajectory Prediction Models

Pedestrian trajectory prediction models were trained using the official code for each model. Adam optimizer and multi-step learning rate scheduler were used. Social-LSTM [1], Y-Net [9], and NSP [14] were trained for 20 epochs with a learning rate set to $0.003, 0.001, 0.001$, and batch sizes of $5, 4$, and $8$. NSP used the results of the model trained with Y-Net as goal point information. When using semantic map information, we used $0.08m$ per pixel as a patch size.

### B.4 End-to-End Motion Forecasting Models

For the end-to-end pedestrian motion forecasting task, we implemented *Fast and Furious* (FaF) [8] and FutureDet [10] using the PyTorch 1.7.1 framework and the 1.0 version of MMDetection3D [4]. The batch sizes were set to 2 and 8 respectively. We trained each model for 20 epochs using the AdamW Optimizer with a learning rate of 0.001. Both FaF and FutureDet were trained using the code from the FutureDet author's repository.

## C Performance Metrics

### C.1 3D Pedestrian Detection

***Average Precision* (AP)**

$$\text{AP}_d = \sum_n (R_n - R_{n-1}) P_n, \tag{1}$$

where $R_n$ and $P_n$ are the recall and precision at the $n$th confidence score threshold and $d$ denotes distance threshold.

***mean Average Precision* (AP)**

$$\text{mAP} = \frac{1}{4} \sum_d \text{AP}_d, \tag{2}$$

where distance threshold is used at $0.25, 0.5, 1$, and $2m$.

### C.2 3D Multi-Object Tracking

***Multiple Object Tracking Accuracy* (MOTA)**

$$\text{MOTA} = 1 - \frac{\sum_t (\text{FP}_t + \text{FN}_t + \text{IDS}_t)}{\sum_t \text{GT}_t}, \tag{3}$$

where $\text{FP}_t$, $\text{FN}_t$, $\text{IDS}_t$, and $\text{GT}_t$ denotes number of False Positive, False Negative, IDentity Switches and Ground Truth objects at time $t$, respectively.

*Multiple Object Tracking Precision* (**MOTP**)

$$\text{MOTP} = \frac{\sum_t \sum_i d_{t,i}}{\sum_t c_t}, \tag{4}$$

where $d_{t,i}$ denotes the distance between the predicted bounding box $i$ and its corresponding ground truth bounding box at time $t$, and $c_t$ denotes the number of matches correctly identified objects at time $t$.

*Average Multiple Object Tracking Accuracy* (**AMOTA**)

$$\text{AMOTA} = \frac{1}{N} \sum_{i=1}^{N} \text{MOTA}_{r_i}, \tag{5}$$

where $r_i$ denotes $i$th recall value and $N$ is the number of recall values.

*Average Multiple Object Tracking Precision* (**AMOTP**)

$$\text{AMOTP} = \frac{1}{N} \sum_{i=1}^{N} \text{MOTP}_{r_i} \tag{6}$$

*scaled MOTA* (**sMOTA**)

$$\text{sMOTA}_r = max(0, 1 - \frac{\text{FP}_r + \text{FN}_r + \text{IDS}_r - (1 - r) \times \text{GT}}{r \times \text{GT}}) \tag{7}$$

*scaled AMOTA* (**sAMOTA**)

$$\text{sAMOTA} = \frac{1}{N} \sum_{i=1}^{N} \text{sMOTA}_{r_i} \tag{8}$$

## C.3 Pedestrian Trajectory Prediction

*Average Displacement Error* (**ADE**)

$$\text{ADE} = \frac{1}{T} \sum_{t=1}^{T} ||y_t - \hat{y}_t||, \tag{9}$$

where $T$ represents the prediction horizon, $y_t$ and $\hat{y}_t$ denote the ground truth and the predicted position at time step $t$, respectively.

*Final Displacement Error* (**FDE**)

$$\text{FDE} = ||y_T - \hat{y}_T|| \tag{10}$$

*Minimum Average Displacement Error* (**ADE$_K$**)

$$\text{ADE}_K = \min_{i=1:K} \text{ADE}_i, \tag{11}$$

where $K$ denotes the number of predicted trajectories.

*Minimum Final Displacement Error* (**FDE$_K$**)

$$\text{FDE}_K = \min_{i=1:K} \text{FDE}_i \tag{12}$$

## C.4 End-to-End Pedestrian Motion Forecasting

*Forecasting Mean Average Precision* (**mAP$_f$**)

$$\text{mAP}_f = \frac{1}{3}(\text{AP}_f^S + \text{AP}_f^L + \text{AP}_f^{NL}), \tag{13}$$

where $\text{AP}_f$ represents mAP in the predicted final timestep, while $S$, $L$, and $NL$ denote static, linear, and non-linear states defined by FutureDet [10].