# OpenReview forum: "SiT Dataset: Socially Interactive Pedestrian Trajectory Dataset for Social Navigation Robots"
_NeurIPS.cc/2023/Track/Datasets_and_Benchmarks — NeurIPS 2023 Datasets and Benchmarks Poster_

### Official Review · Reviewer_RCVX · 2023-07-20
**SiT Dataset: Socially Interactive Pedestrian Trajectory Dataset for Social Navigation Robots**

**Rating:** 7
**Confidence:** 4
**Clarity:** Yes, the paper is well-written and ea…

**Strengths:**

It provides a valuable resource for developing perception and motion prediction models for social navigation robots. The paper is well-organized and provides a thorough evaluation of the dataset. The SiT dataset has the potential to open up research opportunities in the field of social navigation robots and enable the development of more effective social navigation robots.

**Additional Feedback:**

We know that crucial challenges are posed by occlusions and close proximity among pedestrians in highly crowded areas. I would like to inquire whether these issues have been effectively addressed in the annotations of your dataset. Additionally, I am interested to know whether these annotations have proven helpful for the models in tackling these specific issues during the trajectory detection, tracking, and prediction tasks.

**Correctness:**

The dataset is documented and structured in a standardized manner. The benchmarks were assessed using conventional metrics.

**Documentation:**

The paper is well-organized and clearly presents the authors' research and findings. The authors have provided detailed descriptions of their methods and results, and have included appropriate references to related work in the field.

**Ethics:**

As the authors said, all images have been processed to blur pedestrian faces and license plates, and all information is only used for research purposes.

**Limitations:**

The authors have addressed potential negative societal impacts. As for the dataset size limitation, the authors promise they will enlarge the size in the future V2 version.

**Opportunities For Improvement:**

1. Limited interaction between pedestrians and robots: The dataset primarily captures pedestrians passing by without significant interaction with the robot, potentially limiting the representation of social navigation challenges.
2. Inadequate long-term trajectories: The dataset might lack long-term pedestrian trajectories, which are important for understanding behavior patterns in various environments sometimes.

**Relation To Prior Work:**

Yes, this paper thoroughly discusses related work, such as Waymo, JRDB, STCrowd, etc., and identifies differences among them.

**Summary And Contributions:**

The paper introduces a new dataset of pedestrian trajectories with social navigation robots, which is designed to facilitate the development of 3D pedestrian detection, tracking, and trajectory prediction. The main contributions of this dataset are:
1. Provide a large-scale real-world pedestrian trajectory dataset in a robot view in densely populated indoor and outdoor environments.
2. Flexibility to design trajectory prediction models using various contextual information such as appearance features, robot's ego-motion, and semantic map, which could not be supported in the existing datasets.
3. Semantic map data for both indoor and outdoor scenes.

---

> ### Author Response · Authors · 2023-08-21
>
> Dear Reviewer RCVX,
>
> We appreciate your thorough review of our paper.  We aim to address your concerns in our subsequent responses. Additionally, we've incorporated supplementary materials in the updated supplementary material to support our clarifications.
>
> **1. Limited Interaction Between Pedestrians and Robots:**
>
> In our SiT dataset, we made our best efforts to collect realistic trajectory data that reflects natural pedestrian behavior and interactions in the real world. As in Fig. 8 in the revised supplementary material, our dataset includes various interactive scenarios reflecting real-world navigation like approaching, following, collision avoidance, etc. Although our dataset could not cover all human-robot interaction cases and active interactions like collaborative works, we believe that the SiT dataset captures scenarios pertinent to social navigation in congested and cluttered environments. We acknowledge that there may be room for further improvement in this aspect, and we will consider including more diverse and complex interactions in future updates of the dataset.
>
>
> **2. Inadequate Long-Term Trajectories:**
>
> Your point about the importance of long-term pedestrian trajectories is well taken. In the SiT dataset, we thought that a 7-second trajectory data span is adequate for developing prediction and planning modules for social navigation robots. However, we recognize the value of long-term trajectories in understanding complex pedestrian behavioral patterns, and we will continue to explore ways to include more of these in our dataset to increase its applicability and richness.
>
>
> **3. Challenges Posed by Occlusions and Close Proximity:**
>
> We also encountered challenges in labeling objects in 3D due to lack of data in case of occlusions and the close proximity of pedestrians in densely crowded areas. We enhanced the labeling accuracy of occluded objects by cross-referencing both the camera and LiDAR data, and by referencing labeling results across consecutive frames. We leveraged a 3D labeling software tool developed for this task. We also tried to annotate nearly all objects in close proximity, as long as they were discernible to humans. We have not yet conducted a comprehensive analysis of the impact of our labeling strategy on performance, but we will continue to investigate this in the future.

---

### Official Review · Reviewer_qffA · 2023-07-21
**Review for SiT Dataset**

**Rating:** 7
**Confidence:** 5
**Clarity:** The paper is generally written-well.

**Strengths:**

1. This paper effectively addresses the research gap in pedestrian trajectory forecasting by introducing a valuable dataset of social navigation robot scenarios. The unique setting, where the robot is in motion while pedestrians are widely dispersed (as shown in Fig 7.), makes it distinct from common autonomous driving datasets like Waymo and Argoverse or surveillance video datasets like ETH/UCY. The significance of social navigation robots in trajectory forecasting and tracking systems further underscores the importance of this dataset. I am glad to see the authors collect and annotate data for this specific scenario.

2. In addition to its strong motivation, the dataset exhibits high-quality characteristics. The inclusion of multi-modal sensors, diverse locations, and semantic maps aligns well with the research community's interests. The authors' provision of benchmark results enhances the dataset's potential impact on future research in related areas.




**Additional Feedback:**

N/A

**Correctness:**

Although this reviewer may miss some details, the claims in the paper are technically correct to this reviewer.

**Documentation:**

The URL contains access to the dataset and the license. The paper mentions the maintenance plan. The appendix reveals the implement details of the benchmark.

**Ethics:**

Since the dataset collects real-world data from real people, this reviewer is not sure if the authors should ask for the explicit consent of participants.

**Limitations:**

1. This paper contains real-world data. This reviewer may concern about the potential ethical issues during the data collection.

2. The authors mentioned it is hard to scale the size of the dataset up to hundreds of thousands of frames due to the time constraints but they would add more data in the future.

**Opportunities For Improvement:**

As the dataset primarily centers around pedestrian detection/prediction, I'm curious if the authors have any intentions to incorporate other visual annotations, such as human poses. Including human poses in the dataset could be beneficial for action analysis and motion prediction, while also offering valuable insights for social navigation robot scenarios.

**Relation To Prior Work:**

Yes. Since this dataset introduces an overlooked scenario to the trajectory forecasting community, the paper has discussed the differences between the proposed dataset and the existing ones.

**Summary And Contributions:**

This paper proposed a pedestrian trajectory dataset for social navigation robots. It contains a variety of scenes of indoor/outdoor environments. The dataset has rich annotations which support three crucial tasks for social navigation robots: detection, tracking, and prediction. The multi-modal, mobile sensor setting is also different from the existing popular datasets for the pedestrian trajectory prediction task.

---

> ### Author Response · Authors · 2023-08-21
>
> Dear Reviewer qffA,
>
> We appreciate your thorough review of our paper.  We aim to address your concerns in our subsequent responses. Additionally, we've incorporated supplementary materials in the updated Supplementary Material to support our clarifications.
>
> **1. Consider Additional Labeling:**
>
> We thank the reviewer for the suggestion. We agree with the reviewer that adding further visual annotations, like human poses, would enhance the value of this dataset. We also think these pose annotations can be utilized to explore the task of identifying and predicting human actions based on sequential pose data. We will certainly consider it as we continue to develop and expand our work.
>
>
> **2. Potential Ethical Issues During the Data Collection Process:**
>
> We understand and share your concerns about the ethical considerations involved in collecting real-world data. Given time and manpower limitations, we were unable to secure consent for data collection from pedestrians and vehicles. Under current Korean legislation, data can be utilized for research purposes without consent, provided it has been properly de-identified. To comply with this, we blurred human faces and vehicle license plates to anonymize the data, as described in supplementary material A.5. Our labeling team conducted a thorough triple-check to ensure privacy guidelines. Moreover, we demanded confidentiality agreements during the data processing and labeling stage to prevent potential data leaks before the process of de-identifying the data.

---

### Official Review · Reviewer_YC2B · 2023-07-21
**SiT Dataset is a potential application area in the future.**

**Rating:** 6
**Confidence:** 4
**Correctness:** yes
**Clarity:** yes

**Strengths:**

The SiT dataset significantly advances the field of social navigation robots by providing a rich resource for training and evaluating models that perceive and predict pedestrian trajectories. Unlike existing datasets that are based on static scenes, SiT captures the dynamic human-robot interactive scenarios in motion, better mimicking real-world applications.

**Additional Feedback:**

no

**Documentation:**

yes

**Limitations:**

The authors have outlined their work, but they have not explicitly addressed all of its potential limitations.

**Opportunities For Improvement:**

1. The practicality of the SiT dataset is also noteworthy consideration. For instance, if the environment or situation during data collection is overly simplistic or idealized compared to the complexity of the real world, the performance of models trained using this dataset might be limited in real-world applications.
2. Ethical and Privacy Considerations. While the authors have taken steps to anonymize data, the inherent nature of the dataset (tracking and predicting pedestrian movement) may have privacy implications that need to be further addressed.
3. Beyond pedestrian trajectory prediction for robot navigation, the dataset could potentially be utilized in various other areas. For example, the rich information about pedestrian behaviors could be leveraged in behavioral studies or urban planning to understand how people move and interact in different environments. Moreover, the dataset could be useful in the development of intelligent transportation systems, enhancing autonomous vehicle technologies by improving pedestrian detection and prediction.

**Relation To Prior Work:**

yes

**Summary And Contributions:**

The SiT dataset contributes to the field by providing a robust foundation for future research in pedestrian trajectory prediction. It brings the possibility to accelerate the development of safe and agile social navigation robots. The dataset, development toolkit, and pre-trained models are publicly available.

---

> ### Author Response · Authors · 2023-08-21
>
> Dear Reviewer YC2B,
>
> We appreciate your thorough review of our paper.  We aim to address your concerns in our subsequent responses. Additionally, we've incorporated supplementary materials in the updated supplementary material to support our clarifications.
>
> **1. Practicality of the SiT Dataset:**
>
> We appreciate your concerns regarding the practicality of the SiT dataset in reflecting the complexity of the real world. We made our best efforts to collect realistic trajectory data that reflects natural pedestrian behavior and interactions:
>
> Real-world dataset: Our dataset emphasizes on capturing natural and unbiased interactions between pedestrians and the robot in real-world scenarios. We refrained from using controlled setups or providing guidance to participants, instead gathering trajectory data from pedestrians encountered in both indoor and outdoor real-world settings.  As in Fig. 8 in the revised supplementary material, our dataset includes various interactive scenarios reflecting real-world navigation like approaching, following, collision avoidance, etc. Our main interest lies in capturing interactions related to navigation in crowded and cluttered areas, and in this regard, we believe our dataset fulfills this objective.
>
>
> Diverse environments: The data was collected in various indoor and outdoor locations, including corridors, cafeterias, university buildings, streets, and pedestrian crossings. Also, the dataset included scenarios with varying the number of pedestrians per location (shown in the table below). This diversity ensures that the dataset encompasses a broad spectrum of real-world situations that a self-driving robot might encounter during social navigation.
>
> |       **Location**        | **In/Out** | **# of Objects per Scene** |
> |:-------------------------:|:---------:|:--------------------------:|
> |        Cafe Street        |  Outdoor  |            12.4k           |
> | Three-way Intersection    |  Outdoor  |             6.4k           |
> |       Outdoor Alley       |  Outdoor  |            15.9k           |
> |        Crossroad          |  Outdoor  |            15.0k           |
> |     Subway Entrance       |  Outdoor  |             6.3k           |
> |        Courtyard          |  Outdoor  |             4.1k           |
> |         Hallway           |   Indoor  |             3.1k           |
> |        Corridor           |   Indoor  |             2.8k           |
> |       Cafeteria           |   Indoor  |             5.7k           |
> |         Lobby             |   Indoor  |             7.7k           |
> |        **Average**        |     -     |           **7.9k**         |
>
>
>
> **2. Ethical and Privacy Considerations:**
>
> We appreciate your concerns about the ethical and privacy aspects of our SiT dataset. It is a priority for us to ensure privacy and ethical integrity in the collection, processing and distribution of data.
>
> Before distributing the data, we took confidentiality pledges during the data processing/labeling phase to mitigate potential data leakage prior to the de-identification process. As described in supplementary material A.5, we anonymized the data by blurring human faces and vehicle license plates, and conducted a thorough triple check by our labeling team to ensure privacy. However, although we've blurred faces and license plates, complete de-identification remains a challenge due to various visual patterns, such as clothing, gait or movement, that may reveal someone's identity. To address these concerns, we will offer individuals the option to 'Request Takedown Privacy' if they feel their privacy is at risk.
>
> **3. Utilization in other Areas:**
>
> Your insight into the potential applications of the SiT dataset in diverse fields is very valuable. We believe that the diverse and rich information contained in the dataset can be of great value in many fields. In our future work and collaborations, we expect to explore these broader uses and encourage researchers in a variety of fields to use the SiT dataset.
>
> **4. Potential Limitations:**
>
> We acknowledge that certain features, such as pedestrian motion, that could not be anonymized, can cause the potential privacy implications. Thank you for pointing this out, and we will comment on the potential limitations and strategies to mitigate them in the final version.

---

> > ### Comment · Area_Chair_SNLp · 2023-08-28
> >
> > Dear Reviewer YC2B,
> >
> > Please review the author's rebuttal and other feedback to determine if your concerns have been addressed. We appreciate your input. Thank you.
> >
> > Best, AC

---

### Official Review · Reviewer_b9eM · 2023-07-21
**A Valuable Dataset for Socially Compliant Robot Navigation**

**Rating:** 7
**Confidence:** 3
**Clarity:** The exposition is written very clearly.

**Strengths:**

The authors comprehensively justify the need for their dataset, provide a thorough analysis of the dataset's scenes, and report the performance of baseline models on their benchmark.

**Additional Feedback:**

N/A

**Correctness:**

While it has been noted that ETH, UCY, and SDD do not directly cater to motion prediction for socially compliant robot navigation, human-human interaction is nonetheless important to consider for robot navigation. Furthermore, since the attitudes that people have towards robots has an impact on HRI, human-only datasets like those aforementioned are representative of a totally agnostic attitude towards robots, which still have value.

**Documentation:**

There are sufficient details about the data collection, licensing, and availability.
Sufficient details/code have been provided to support reproducibility for their benchmarking.

**Ethics:**

The authors have blurred both license plates and people's faces in their collected image data.

The authors have noted in the checklist that they have discussed consent, but I cannot find where in either the main or supplementary texts. Was ethics approval or a permit obtained to deploy the robot in public spaces in close proximity to pedestrians?

**Limitations:**

For the prediction task, I believe there needs to be a stronger justification against HTP datasets. The fixed perspective is not limiting within the scope of the prediction task and these datasets provide valuable higher-density scenarios for learning to predict human trajectories.

**Opportunities For Improvement:**

Why has the analysis for vehicle datasets like Waymo, argoverse, and nuScenes (Figures 7 and 8) not included ETH, UCY, and SDD?
The stark difference between SiT and Waymo/nuScenes with respect to forward and horizontal difference is due to the ego vehicle driving in lanes. A more fair comparison would be to with an HTP dataset, where heading can be computed from velocity.
This is also apparent in Figure 8's normalized distance density.

It would be helpful to know how many hours of data were collected in total and how many on average per scene. From what I can tell, it seems that there are 20 minutes of data recorded per scene. Is this correct? On page 3 line 64, it says 20 seconds of data were collected and that there are 9 seconds of trajectory data.

**Relation To Prior Work:**

The authors have made numerous comparisons between their dataset and existing datasets.

**Summary And Contributions:**

The authors propose the Social Interactive Trajectory dataset for detecting, tracking, and predicting pedestrian trajectories for socially compliant robot navigation. This dataset was collected as a robot moves through a crowded environment. The authors also provide a benchmark for the tasks and provide baselines.

---

> ### Author Response · Authors · 2023-08-21
>
> Dear Reviewer b9eM,
>
> We appreciate your thorough review of our paper.  We aim to address your concerns in our subsequent responses. Additionally, we've incorporated supplementary materials in the updated supplementary material to support our clarifications.
>
> **1. Comparison with Human Trajectory Prediction Datasets:**
>
> As you suggested, we additionally conducted a comparative analysis with Human Trajectory Prediction (HTP) datasets like ETH/UCY and SDD. As depicted in Fig. 1 of the updated in supplementary material, the SiT dataset exhibits different spatial distribution patterns of pedestrians compared to the ETH/UCY and SDD datasets. In the SiT dataset, the distance between pedestrians is greater than the distance observed between pedestrians in the ETH/UCY and SDD datasets. Additionally, as shown in Fig. 2 in the modified supplementary material, the SiT dataset accommodates more pedestrians than other HTP datasets at every 5m intervals. This seems attributed to the dynamic nature of the scenes captured as the robot moves. In a nutshell, we believe that our SiT dataset exhibits unique interaction behaviors compared to other HTP datasets, owing to the fact that our data is gathered through a mobile robot's interactions with pedestrians in its vicinity.
>
> **2. Details of Data Collection:**
>
> In our dataset, we initially collected approximately three hours of data in the downtown area of Seoul, South Korea, selecting a variety of indoor and outdoor locations including hallways, cafeterias, university buildings, streets, and crosswalks. From this extensive collection, we carefully chose 60 scenes, each lasting 20 seconds, from 10 distinct locations (4 indoor and 6 outdoor). Finally, complete 9-second trajectories are extracted for all pedestrian agents existing in each 20-second scene. A detailed breakdown of the distribution across the different scenes is shown in the table below.
>
> |        **Location**        | **In/Out** | **# of Scenes** | **Duration** | **# of 3d Boxes** | **# of Trajectories** | **# of 2d Boxes** |
> |:--------------------------:|:----------:|:---------------:|:------------:|:-----------------:|:---------------------:|:-----------------:|
> |           Cafe Street      |  Outdoor   |       5         |     100s     |        62k        |          19k          |        68k        |
> | Three-way Intersection     |  Outdoor   |       5         |     100s     |        32k        |           7k          |        45k        |
> |         Outdoor Alley      |  Outdoor   |       4         |     80s      |        63k        |          20k          |        62k        |
> |          Crossroad         |  Outdoor   |       2         |     40s      |        30k        |          12k          |        31k        |
> |       Subway Entrance      |  Outdoor   |       5         |     100s     |        31k        |          11k          |        28k        |
> |          Courtyard         |  Outdoor   |       9         |     180s     |        36k        |          13k          |        33k        |
> |           Hallway          |   Indoor   |       6         |     120s     |        19k        |           6k          |        18k        |
> |           Corridor         |   Indoor   |      11         |     220s     |        31k        |          10k          |        34k        |
> |          Cafeteria         |   Indoor   |       4         |     80s      |        23k        |           4k          |        23k        |
> |           Lobby            |   Indoor   |       9         |     180s     |        70k        |          16k          |        64k        |
> |          **Total**         |      -     |     **60**      |   **20m**    |      **400k**      |        **120k**       |      **500k**     |
>
>
> **3. Ethics Approval to Deploy the Robot in Public Spaces:**
>
> We received approval from the local traffic safety authorities to drive the robot alongside pedestrians on sidewalks to collect datasets, and we ensured that the robot controller was always close by to avoid any harm to pedestrians. For safety, we also set the robot's operating speed to no more than 1 m/s  and equipped the robot with an emergency braking system.

---

### Official Review · Reviewer_UimK · 2023-08-05
**Review of SiT Dataset**

**Rating:** 6
**Confidence:** 4
**Correctness:** Yes.
**Clarity:** The paper is generally well-written.

**Strengths:**

The dataset consists of real-world pedestrian trajectory data **collected by a real robot** navigating both indoor and outdoor environments.
It also includes semantic map data, providing flexibility for designing trajectory prediction models.


**Additional Feedback:**

N/A.

**Documentation:**

Yes, it looks promising.

**Ethics:**

Check Limitations.

**Limitations:**

L294-297 discuss the challenges of scaling up the dataset. It would be beneficial to further explore limitations, such as privacy concerns and biases in real-world data collection.

**Opportunities For Improvement:**

Section 4 Data Analysis does not include detailed data analysis of the length of the collected data and its distribution across different scenes.
The paper does not capture more fine-grained human interactions with the robot, limiting its potential applications.

**Relation To Prior Work:**

Yes, previous contributions have been thoroughly discussed.

**Summary And Contributions:**

The paper introduces the SiT dataset, which consists of extensive pedestrian trajectory data collected by a robot navigating both indoor and outdoor environments. This dataset includes semantic map information and allows for flexible design of trajectory prediction models. The contributions of this work include accurate time synchronization between sensors and comprehensive semantic map data.

---

> ### Author Response · Authors · 2023-08-21
>
> Dear Reviewer UimK,
>
> We appreciate your thorough review of our paper.  We aim to address your concerns in our subsequent responses. Additionally, we've incorporated supplementary materials in the updated supplementary material to support our clarifications.
>
>
> **1. Length of Data Collected and its Distribution over Different Scenes:**
>
> We appreciate the reviewer's suggestion to conduct a detailed analysis of the data length and distribution. In our dataset, we initially collected approximately three hours of data in the downtown area of Seoul, South Korea, selecting a variety of indoor and outdoor locations including hallways, cafeterias, university buildings, streets, and crosswalks. From this extensive collection, we carefully chose 60 scenes, each lasting 20 seconds, from 10 distinct locations (4 indoor and 6 outdoor). The distribution over different scenes is described in the table below:
>
> |        **Location**        | **In/Out** | **# of Scenes** | **Duration** | **# of 3d Boxes** | **# of Trajectories** | **# of 2d Boxes** |
> |:--------------------------:|:----------:|:---------------:|:------------:|:-----------------:|:---------------------:|:-----------------:|
> |           Cafe Street      |  Outdoor   |       5         |     100s     |        62k        |          19k          |        68k        |
> | Three-way Intersection     |  Outdoor   |       5         |     100s     |        32k        |           7k          |        45k        |
> |         Outdoor Alley      |  Outdoor   |       4         |     80s      |        63k        |          20k          |        62k        |
> |          Crossroad         |  Outdoor   |       2         |     40s      |        30k        |          12k          |        31k        |
> |       Subway Entrance      |  Outdoor   |       5         |     100s     |        31k        |          11k          |        28k        |
> |          Courtyard         |  Outdoor   |       9         |     180s     |        36k        |          13k          |        33k        |
> |           Hallway          |   Indoor   |       6         |     120s     |        19k        |           6k          |        18k        |
> |           Corridor         |   Indoor   |      11         |     220s     |        31k        |          10k          |        34k        |
> |          Cafeteria         |   Indoor   |       4         |     80s      |        23k        |           4k          |        23k        |
> |           Lobby            |   Indoor   |       9         |     180s     |        70k        |          16k          |        64k        |
> |          **Total**         |      -     |     **60**      |   **20m**    |      **400k**      |        **120k**       |      **500k**     |
>
>
>
>    We acknowledge the importance of this detailed distribution and will add this information in the revised version of our paper.
>
> **2. Human Interactions with the Robot:**
>
>   Our dataset emphasizes on capturing natural and unbiased interactions between pedestrians and the robot in real-world scenarios. We avoided controlled setups and did not provide any guidance to participants, collecting trajectory data from pedestrians in both indoor and outdoor real-world environments. As in Fig. 8 in the revised supplementary material, our dataset includes various interactive scenarios reflecting real-world navigation like approaching, following, collision avoidance, etc. Although our dataset could not cover active human-robot interactions like dialogue or collaborative works, our main interest lies in capturing interactions related to navigation in crowded and cluttered areas, and in this regard, we believe that our dataset fulfills this objective.
>
>
> **3. Privacy Considerations:**
>
> We appreciate your concerns about the privacy aspects of our SiT dataset. It is a priority for us to ensure privacy and ethical integrity in the collection, processing and distribution of data.
>
> Before distributing the data, we took confidentiality pledges during the data processing/labeling phase to mitigate potential data leakage prior to the de-identification process. As described in supplementary material A.5, we anonymized the data by blurring human faces and vehicle license plates, and conducted a thorough triple check by our labeling team to ensure privacy. However, although we've blurred faces and license plates, complete de-identification remains a challenge due to various visual patterns, such as clothing, gait or movement, that may reveal someone's identity. To address these concerns, we will offer individuals the option to 'Request Takedown Privacy' if they feel their privacy is at risk.

---

> > ### Comment · Reviewer_UimK · 2023-08-22
> >
> > Thank you for addressing my questions and concerns. I appreciate your hard work on this project. I will maintain my positive rating and look forward to seeing more progress in expanding and leveraging the dataset.

---

### Decision · Program_Chairs · 2023-09-22

**Decision:**

Accept (Poster)

**Comment:**

This work introduces the Social Interactive Trajectory dataset, designed for detecting, tracking, and predicting pedestrian trajectories in real-world scenarios. Collected by a robot navigating diverse indoor and outdoor environments, the dataset also incorporates a benchmark for related tasks and offers baseline measurements.

In the rebuttal process, several reviewers raised several concerns, including privacy consideration. The authors have adequately addressed these issues. Consequently, all reviewers agree to accept.